# L-CAD: Language-based Colorization with Any-level Descriptions using Diffusion Priors

**Zheng Chang**[#1]    **Shuchen Weng**[#2,3]    **Peixuan Zhang**[1]    **Yu Li**[4]    **Si Li**[*1]    **Boxin Shi**[2,3]

[1] School of Artificial Intelligence, Beijing University of Posts and Telecommunications
[2] National Key Laboratory for Multimedia Information Processing
School of Computer Science, Peking University
[3] National Engineering Research Center of Visual Technology
School of Computer Science, Peking University
[4] International Digital Economy Academy
{zhengchang98,pxzhang,lisi}@bupt.edu.cn
{shuchenweng, shiboxin}@pku.edu.cn    liyu@idea.edu.cn

## Abstract

Language-based colorization produces plausible and visually pleasing colors under the guidance of user-friendly natural language descriptions. Previous methods implicitly assume that users provide comprehensive color descriptions for most of the objects in the image, which leads to suboptimal performance. In this paper, we propose a unified model to perform language-based colorization with any-level descriptions. We leverage the pretrained cross-modality generative model for its robust language understanding and rich color priors to handle the inherent ambiguity of any-level descriptions. We further design modules to align with input conditions to preserve local spatial structures and prevent the ghosting effect. With the proposed novel sampling strategy, our model achieves instance-aware colorization in diverse and complex scenarios. Extensive experimental results demonstrate our advantages of effectively handling any-level descriptions and outperforming both language-based and automatic colorization methods. The code and pretrained models are available at: *https://github.com/changzheng123/L-CAD*.

## 1   Introduction

Image colorization is a challenging task that involves converting grayscale images into plausible and visually pleasing colorful ones. Language-based colorization approaches (*e.g.*, [5, 6, 27]) use natural language descriptions as guidance to produce more controllable colorized images. These methods cater to users' specific requests, enabling them to provide more specific and nuanced color preferences. As automatic colorization (*e.g.*, [39, 44, 52]) often encounters color ambiguity for common objects (*e.g.*, flower colors), language-based colorization has shown promising results in producing high-quality and customizable colorized images with user-friendly input.

While language-based colorization methods have improved the consistency between language descriptions and colorization results through feature fusion [7, 27], decoupling color-object space [5, 45], and aggregating similar patches [6], they implicitly assume that users provide comprehensive color descriptions for most of the objects in the image. This assumption often leads to suboptimal performance, especially for objects without corresponding color descriptions. Furthermore, we have observed that users commonly assign colors only to their objects of interest.

---

# Equal contribution. * Corresponding author

37th Conference on Neural Information Processing Systems (NeurIPS 2023).

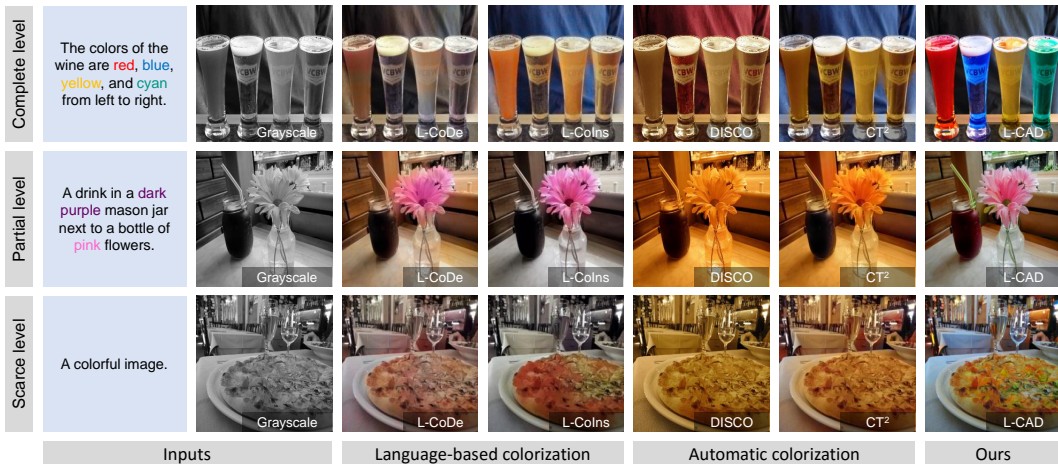

Figure 1: Colorization with any-level descriptions, comparing language-based colorization methods (*e.g.*, L-CoDe [45] and L-CoIns [6]) and automatic colorization methods (*e.g.*, DISCO [47] and CT² [44]). **First row:** For complete-level descriptions, automatic colorization methods is unable to colorize images as requested by the user. **Second row:** For partial-level descriptions, automatic colorization methods fail to colorize mentioned objects with their corresponding colors, while language-based colorization methods struggle to present appropriate colors for unmentioned objects. **Third row:** For scarce-level descriptions, language-based colorization methods show suboptimal performance without meaningful color hints. In contrast, our method is able to colorize images with any-level descriptions.

To colorize images with descriptions of diverse levels of details, we propose a unified model that adaptively understands **any-level descriptions** and operates as follows: *(i)* For complete-level descriptions that cover major objects, the model precisely colorizes the image based on user requests (Fig. 1 first row, comprehensive descriptions for four cups of wines); *(ii)* for partial-level descriptions that focus on objects of interest, it colorizes unmentioned objects by cuing from image semantics when concentrating on their objects of interest (Fig. 1 second row, selective descriptions for only the jar and flowers); and *(iii)* for scarce-level descriptions that lack meaningful color information, it switches to the automatic colorization model (Fig. 1 third row, omitted descriptions for the pizza and restaurant).

To achieve the above goal, we present **L-CAD** to perform **L**anguage-based **C**olorization with **A**ny-level **D**escriptions. Given the inherent ambiguity in the number of objects mentioned in any-level descriptions, we leverage the pretrained cross-modality generative model (*i.e.*, Stable Diffusion [34]) to utilize its robust language understanding for mentioned objects and rich color priors for unmentioned ones. However, as the generative model is not specifically designed for colorization, it faces challenges in spatial alignment with grayscale inputs. To address this, we develop a luminance-guided compression module that aligns with grayscale images in the pixel space, preserving local spatial structures. We also design a channel-extended convolution operator to prevent the ghosting effect by aligning with descriptions in the latent space. Additionally, to handle descriptions with diverse levels of details and complex scenarios, we develop an instance-aware sampling strategy that roughly estimates object contours in the latent space and assigns color features to their corresponding regions. Our approach is not limited by the number of objects mentioned in descriptions, enabling effective handling of any-level descriptions.

Our contributions could be summarized as follows:

- We propose a unified model that adaptively understands any-level descriptions, achieving state-of-the-art performance in both automatic and language-based colorization methods.

- We develop novel modules to preserve local spatial structures and prevent the ghosting effect by aligning with input conditions in both the pixel space and the latent space.

- We present the instance-aware sampling strategy to correctly assign colors to corresponding objects, enabling effective colorization in diverse and complex scenarios.

## 2 Related Works

### 2.1 Language-based colorization

Language-based colorization aims to leverage flexible and user-friendly language descriptions to produce visually pleasing results with descriptions that contain objects and their corresponding colors. Manjunatha *et al.* [27] introduce the FILM pipeline, which employs feature-wise affine transformations to incorporate language descriptions in the grayscale image colorization process. Similarly, Chen *et al.* [7] utilize a recurrent attentive model to spatially fuse image and language features. To gain a deeper understanding of grayscale images, Xie *et al.* [49] use additional segmentation masks and introduce semantic segmentation as a side task. L-CoDe [45] and L-CoDer [5] rely on additionally annotated correspondences between object words and color words for guidance, enabling the model to assign specific color words to image regions. To achieve instance-aware colorization, L-CoIns [6] presents the grouping mechanism to automatically aggregate similar image patches. In this paper, we further explore the effective approach to enhance the robustness of language-based colorization, *e.g.*, assigning vivid colors for objects unmentioned in the description.

### 2.2 Automatic colorization

Automatic colorization aims to estimate the missing chromatic channels in grayscale images, generating diverse, colorful, and plausible results without relying on additional user-provided guidance. Chen *et al.* [8] pioneer the first deep-learning-based automatic colorization approach. With the advent of CNNs, which significantly increases the representation ability of the neural network, researchers have focused on developing adaptive feature extraction modules [10, 17, 23]. Recently, advanced generative models, *e.g.*, VAEs [9], GANs [4, 15, 40, 41], cINNs [2], and Transformers [18, 22, 47], have been employed to better fit real-world color distributions and produce vivid colorization results. Concurrently, other works have explored incorporating various priors to achieve accurate colorization semantics, including segmentation masks [56, 55], detection boxes [39], rich color prior [44, 52], and pretrained generative models [20, 46]. Despite these advances, automatic colorization still faces challenges in handling multi-modal uncertainty. Considering that rich image semantics are encapsulated in language descriptions, we utilize the prior knowledge from the pretrained cross-modality generative model, *i.e.*, Stable Diffusion [34], to alleviate this issue.

### 2.3 Text-driven image editing

Leveraging the flexibility of description representation, text-driven image editing methods enable precise control over image color, style, contour, and high-level semantics. Early works [24, 43, 51] develop text injection modules and design supervisory signals based on the conditional GAN [29]. Recently, with the development of diffusion models [13, 38], researchers show increasing interests in exploiting text-driven image generation [32, 33, 37], as well as diverse image editing tasks, *e.g.*, text-guided image inpainting [34, 48], inversion-based style transfer [54], and text-driven image-to-image translation [19]. To reduce resource consumption, some researchers build models based on pretrained generative models by advanced sampling strategies [28, 30] or finetuning their weights [35, 50]. Although language-based colorization could be considered as a specialized text-driven image editing task, existing methods utilizing pretrained generative models hardly or seldom succeed in instance-aware colorization due to their failure to spatially align with input grayscale images.

## 3 Methodology

In this section, we first provide a brief review of related diffusion models [13, 34, 38] in Sec. 3.1 to make this paper self-contained. Following that, we delve into the details of designed modules that spatially align with input conditions in Secs. 3.2 and 3.3. Next, we present the instance-aware sampling strategy for effective colorization in diverse and complex scenarios in Sec. 3.4. Finally, training details are shown in Sec. 3.5.

### 3.1 Preliminaries

Diffusion models [13, 38] are one type of generative models that learn to represent the distribution of images from a given dataset. During the inference process, these models iteratively refine a

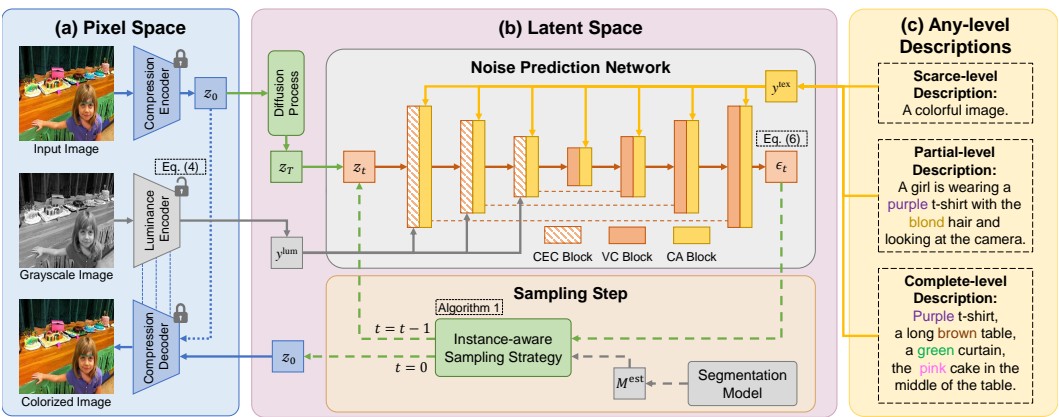

Figure 2: The framework of L-CAD. (a) We introduce an additional luminance encoder to capture multi-scale features from the grayscale image, which preserves local spatial structures of grayscale images and guides the decoding process from the latent space back to pixel space (Sec. 3.2). (b) We modify the noise prediction network by replacing Vanilla Convolution (VC) blocks with our proposed Channel-Extended Convolution (CEC) blocks within each downsampling module before Cross-Attention (CA) blocks. This prevents the ghosting effect by aligning with descriptions in the latent space (Sec. 3.3). For visualizing the colorized image, we further present the instance-aware sampling strategy to correctly assign colors to corresponding objects for descriptions with diverse levels of details and complex scenarios (Sec. 3.4). (c) During training, our model randomly receives any-level descriptions to increase the robustness of colorization (Sec. 3.5).

random noise sample $x_T \sim \mathcal{N}(0,1)$ until it converges to a photorealistic image $x_0$. At each step $t \in \{0, \ldots, T\}$, the intermediate sample $\mathbf{x}_t$ is computed as:

$$x_t = \sqrt{\alpha_t}x_0 + \sqrt{1 - \alpha_t}\epsilon_t, \tag{1}$$

where $\epsilon_t \sim \mathcal{N}(0,1)$ is the Gaussian noise at step $t$. In diffusion models, a neural network denoted as $\epsilon_\theta$ is employed at each step to predict the noise $\epsilon_t$, guiding the model in the process of reversing $x_t$ to $x_{t-1}$ through a denoising procedure.

Diffusion models can be extended to handle conditional generative tasks by incorporating an additional condition $y$. In this case, the noise prediction network $\epsilon_\theta$ takes three parameters as input: the noise sample $x_t$, the step $t$, and the condition $y$. This enables the diffusion process to generate results that are consistent with the given condition. To achieve the denoising objective, diffusion models typically utilize a simple $\mathcal{L}_2$ loss, aiming to minimize the discrepancy between the learned image distribution and the target distribution. The loss function can be formulated as:

$$\mathcal{L}_{\mathrm{dm}} = \mathbb{E}_{t,x_0,\epsilon \sim \mathcal{N}(0,1)} \big[ \| \epsilon_t - \epsilon_\theta(x_t, t, y) \|^2 \big]. \tag{2}$$

To address the challenges of resource consumption and high-resolution image generation, Stable Diffusion [34] introduces a perceptual compression model to make the latent space perceptually equivalent to the image space. This enables the diffusion process to be performed in the compressed latent space. Specifically, it adopts a compression encoder $\mathcal{E}$ to encode the given image $x$ into latent space as $z = \mathcal{E}(x)$, and a compression decoder $\mathcal{D}$ to reconstruct the image as $\tilde{x} = \mathcal{D}(z)$. Notably, the noise prediction network $\epsilon_\theta$ is trained to learn the distribution of the latent representation $z$ instead of the input image $x$.

In this paper, our goal is to leverage Stable Diffusion [34]'s robust language understanding and rich color priors for language-based colorization with any-level descriptions.

## 3.2 Luminance-guided image compression

While Stable Diffusion [34] demonstrates superior performance in text-driven image editing, it lacks the ability to preserve local spatial structures of input grayscale images, which is essential for the colorization task. To address this limitation, we propose the incorporation of an additional luminance

encoder in the pixel space. This encoder serves as a bridge between the colorization results and the grayscale images, ensuring better alignment between the two.

As shown in Fig. 2 (a), the colorful image is encoded into the latent representation by the compression encoder, while the luminance encoder extracts multi-scale features from grayscale images, preserving local structural semantics. These features are added directly into the corresponding scales of the compression decoder, guiding the decoding process from the latent space to the pixel space. The luminance encoder adopts the same architecture as the compression encoder, with the weights of the compression encoder and decoder fixed to retain prior knowledge from the pretrained model.

Inspired by LDL [25], we observe that erroneous pixels in regions with regular and sharp structures significantly damage visual perception. To address this, we intend to estimate an artifact map $M_{h,w}^{\text{art}}$, which would serve to indicate the probability of encountering artifacts at specific spatial position $(h, w)$ within the colorized results. Specifically, we calculate the residual $\delta$ between the ground truth image $x$ and the colorized result $\tilde{x}$. By computing the variance of the residual within local square windows at each position, we generate an artifact map denoted as $M^{\text{art}}$ as:

$$M_{h,w}^{\text{art}} = \sum_{p \in \Omega_{(h,w)}} \left( \frac{\delta_p - \mu_p}{N_{\text{win}}} \right)^2, \qquad \mu_p = \sum_{p \in \Omega_{(h,w)}} \frac{\delta_p}{N_{\text{win}}^2}, \tag{3}$$

where $p$ denotes the position index of the local window $\Omega$ centered on position $(h, w)$ and $N_{\text{win}}$ is the local window size. Given that artifacts typically show up with high-frequency characteristics, areas with higher variances likely indicate where these artifacts are. We could apply the artifact map as a weight to the image reconstruction loss as $\mathcal{L}_{\text{rec}} = \|M^{\text{art}} \odot (x - \tilde{x})\|_1$.

Our total loss to train our model in pixel space is as follows:

$$\mathcal{L}_{\text{pix}} = \mathcal{L}_{\text{rec}} + \alpha \mathcal{L}_{\text{dis}} + \beta \mathcal{L}_{\text{per}}, \tag{4}$$

where $\alpha = 1.0$ and $\beta = 0.5$ are weighting parameters, the discriminator loss $\mathcal{L}_{\text{dis}}$ and the perceptual loss $\mathcal{L}_{\text{per}}$ are adopted to improve the overall quality of reconstructed images.

### 3.3 Semantic-aligned latent representation

In Stable Diffusion [34], descriptions are incorporated into the latent representation using Cross-Attention (CA) blocks to ensure flexibility. However, this approach leads to the ghosting effect as color features are assigned outside the intended regions. To address this, we focus on maintaining semantic alignment between descriptions and grayscale images within the latent space.

In Fig. 2 (b), we replace Vanilla Convolution (VC) blocks in the noise prediction network with our Channel-Extended Convolution (CEC) blocks within each downsampling module. These CEC blocks receive the resized luminance features $y^{\text{lum}}$ from the luminance encoder in the pixel space as additional guidance. By leveraging extended channels, CEC blocks could effectively capture the local structural semantics of the luminance in the latent space. We formulate the channel-extended convolution as:

$$f'_{h,w} = \sum_{i=0}^{N_{\text{k}}-1} \sum_{j=0}^{N_{\text{k}}-1} \left( \sum_{k=1}^{N_{\text{fix}}} \omega_{i,j,k}^{\text{fix}} f_{p,q,k} + \sum_{k=1}^{N_{\text{ext}}} \omega_{i,j,N_{\text{fix}}+k}^{\text{ext}} \bar{y}_{p,q,k}^{\text{lum}} \right), \tag{5}$$

where $h, w, p = h + i - N_{\text{k}}/2$ and $q = w + j - N_{\text{k}}/2$ are position index of feature maps, $f$ and $f'$ are separately feature maps before and after performing the convolution, $N_{\text{k}}$ is the kernel size, $N_{\text{fix}}$ and $N_{\text{ext}}$ separately mean the number of fixed channels and extended channels, $\bar{y}^{\text{lum}}$ denote resized luminance features. Mathematically, the CEC block is equivalent to using a stack of convolutions to extract features and add their output to the downsampling block. To preserve the robust language understanding and rich color priors of the pretrained generative model, we keep the weights $\omega^{\text{fix}}$ fixed during training. Additionally, we initialize the weights of extended channels $\omega^{\text{ext}}$ to zero, ensuring that our model maintains functional equivalence to the pretrained generative model prior to training.

We use skip connections between downsampling and upsampling modules to guide the upsampling process with luminance features. To expedite training convergence, we utilize Channel-Extended Convolution (CEC) blocks only in downsampling modules.

Table 1: Quantitative comparisons with language-based colorization methods (left) and automatic colorization methods (right). ∗ marks our method receiving scarce-level descriptions. Throughout this paper, ↑ (↓) means higher (lower) is better. Best performances are highlighted in **bold**.

| | Comparison with language-based colorization methods | | | | | | Comparison with automatic colorization methods | | | | | |
|---|---|---|---|---|---|---|---|---|---|---|---|---|
| Method | Extended COCO-Stuff | | | Multi-instance | | | Method | Extended COCO-Stuff | | | Multi-instance | | |
| | PSNR↑ | SSIM↑ | LPIPS↓ | PSNR↑ | SSIM↑ | LPIPS↓ | | PSNR↑ | SSIM↑ | LPIPS↓ | PSNR↑ | SSIM↑ | LPIPS↓ |
| LBIE | 22.02 | .8519 | .265 | 21.92 | .8697 | .260 | CIC | 22.03 | .8893 | .232 | 21.95 | .8754 | .239 |
| ML2018 | 21.06 | .8533 | .282 | 20.54 | .8495 | .294 | InstColor | 23.76 | .8979 | .195 | 23.59 | .8968 | .199 |
| Xie2018 | 21.92 | .8764 | .233 | 20.04 | .8530 | .268 | ChromaGAN | 22.08 | .8416 | .275 | 22.41 | .8584 | .248 |
| L-CoDe | 24.86 | .9091 | .166 | 23.84 | .9056 | .176 | BigColor | 21.45 | .8868 | .236 | 21.43 | .8824 | .225 |
| L-CoDer | 25.03 | .9125 | .162 | 24.16 | .9135 | .168 | DISCO | 20.62 | .8744 | .212 | 20.81 | .8762 | .204 |
| L-CoIns | 25.10 | .9151 | .159 | 24.74 | .9141 | .156 | CT$^2$ | 23.16 | .9030 | .191 | 22.65 | .8993 | .193 |
| *W/o* LIC | 23.87 | .8866 | .190 | 23.96 | .9047 | .160 | *W/o* LIC* | 22.25 | .8584 | .206 | 22.53 | .8697 | .176 |
| *W/o* SLR | 17.26 | .5984 | .587 | 15.61 | .4879 | .531 | *W/o* SLR* | 15.73 | .6132 | .512 | 14.18 | .5732 | .472 |
| *W/o* ISS | 25.32 | .9174 | .147 | 24.57 | .9138 | .149 | *W/o* ISS* | 24.24 | .9073 | .159 | 24.07 | .9054 | .165 |
| L-CAD | **25.97** | **.9191** | **.142** | **25.51** | **.9156** | **.127** | L-CAD* | **24.33** | **.9114** | **.157** | **24.19** | **.9108** | **.164** |

The loss for training the denoising network in the latent space is:

$$\mathcal{L}_{\text{lat}} = \mathbb{E}_{t,z_0,\epsilon \sim \mathcal{N}(0,1)} \big[ \| \epsilon_t - \epsilon_\theta(z_t, t, y^{\text{tex}}, y^{\text{lum}}) \|^2 \big], \tag{6}$$

where $y^{\text{tex}}$ is any-level descriptions encoded by the text encoder of CLIP [31].

### 3.4 Instance-aware sampling strategy

To ensure accurate color assignment to objects in images with diverse and complex descriptions, we propose an instance-aware sampling strategy. This strategy draws inspiration from the effectiveness of InstColor [39], which employs an auxiliary recognition model to enhance instance-aware performance. By incorporating the instance-aware sampling strategy into our colorization process, we aim to improve the accuracy and effectiveness of assigning colors to corresponding objects.

To improve instance-awareness, we employ a referring segmentation model (e.g., SAM [21]) to estimate object contours mentioned in the description ($M^{\text{est}}$). Attention maps ($M_l^{\text{att}}$) at the $l$-th CA block control colorized regions for each color word. We normalize the attention maps with Sigmoid and refine them iteratively by aligning with downsampled estimated contours. After that, we apply the denoising process in DDIM [38]. Algorithm 1 outlines our instance-aware sampling strategy.

---

**Algorithm 1:** Instance-aware sampling strategy

**input** : Roughly estimated object contour $M^{\text{est}}$
**output** : Colorized latent representation $z_0$
**for** $t = T \ldots 1$ **do**
  $\_, M_*^{\text{att}} = \epsilon_\theta(z_t, t, y^{\text{lum}}, y^{\text{tex}})$
  **for** $l = 1 \cdots L$ **do**
    $\hat{M}_l^{\text{est}} \leftarrow \text{Downsampling}(M^{\text{est}}, l)$
    $\mathcal{M} \leftarrow \text{Sigmoid}(M_l^{\text{att}})$
    $\hat{M}_l^{\text{att}} \leftarrow M_l^{\text{att}} - \lambda \nabla_{\mathcal{M}} \mathcal{L}_{\text{BCE}}(\mathcal{M}, \hat{M}_l^{\text{est}})$
  **end**
  $\hat{\epsilon}_t, \_ = \epsilon_\theta(z_t, t, y^{\text{lum}}, y^{\text{tex}})\{M_*^{\text{att}} \leftarrow \hat{M}_*^{\text{att}}\}$
  $z_{t-1} = \text{DDIM}(z_t, \hat{\epsilon}_t, t)$
**end**

---

Our instance-aware sampling strategy does not heavily rely on precise segmentation results from the referring segmentation model. This is due to the following reasons: *(i)* Our sampling is performed in the latent space using low-resolution segmentation results, mitigating the impact of segmentation imprecision. *(ii)* When decoding the latent representation into the pixel space, the compression decoder, guided by local structural semantics from grayscale images, could adaptively fix color bleeding issues to produce visually pleasing results.

### 3.5 Learning and implementation details

During training, we utilize the classifier-free guidance [14] to supervise the colorization process conditioned on descriptions. To increase robustness for any-level descriptions, we randomly replace 30% of the complete-level and partial-level descriptions with scarce-level descriptions. The pixel space modules are trained using Eq. (4) with a batch size of 8 for 10 epochs until convergence.

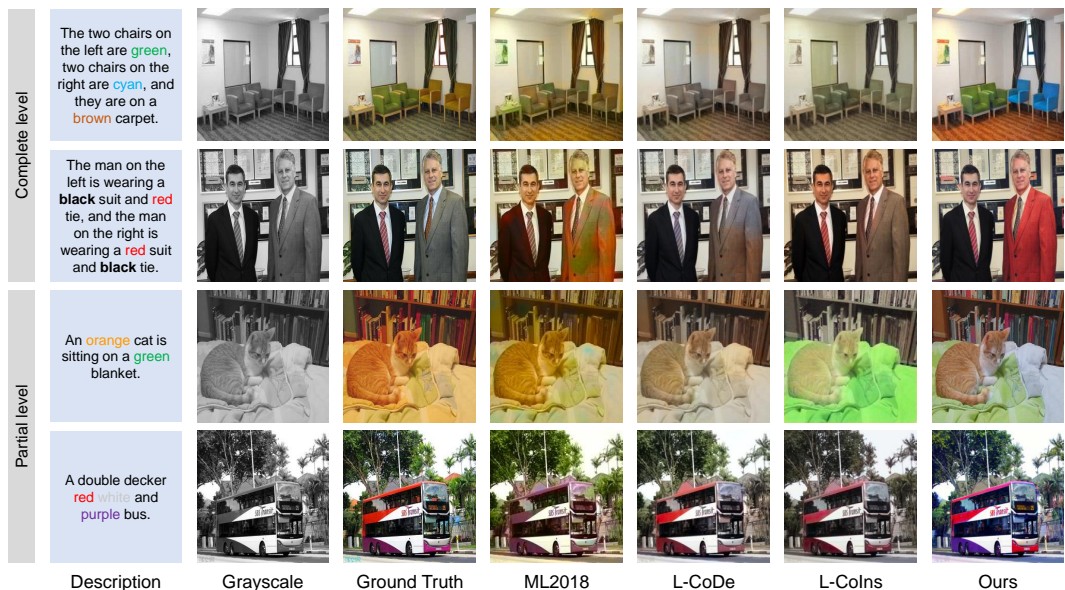

Figure 3: Compared with language-based colorization methods, our model achieves correct instance-aware colorization while improving the colorized quality for unmentioned objects.

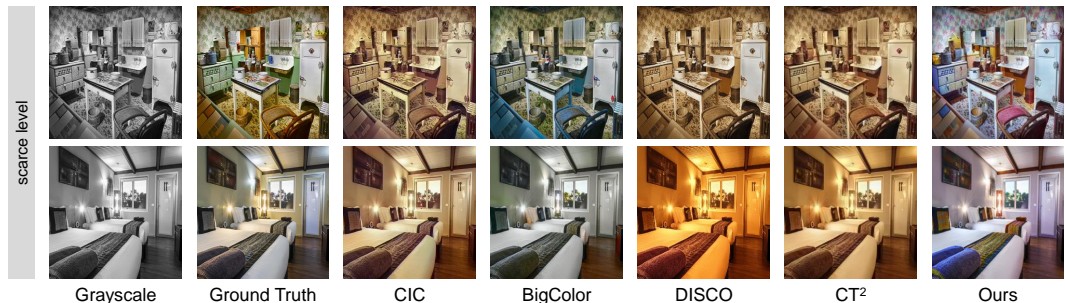

Figure 4: Compared with automatic colorization methods, our model shows comparable performance with scarce-level descriptions that lack meaningful color information.

Subsequently, we fix their weights and train the noise prediction network in the latent space using Eq. (6) for 50 epochs. The training process takes approximately 60 hours and is performed on 2 NVIDIA GeForce 3090Ti graphic cards. We employ the Adam optimizer with a learning rate of $1 \times 10^{-6}$ for the pixel space and $5 \times 10^{-5}$ for the latent space to minimize the losses. For sampling, we use the DDIM sampling method [38] with 50 sampling steps.

## 4    Experiment

**Dataset.** We conduct our experiments on language-based colorization datasets: *(i)* the extended COCO-Stuff dataset [45], which is built upon the COCO-Stuff dataset [3] by discarding unqualified samples for the colorization task, including 59K training images and 2.4K evaluation images; and *(ii)* the multi-instance dataset [6], which provides samples featuring multiple instances with different visual characteristics within a single image, including 65K training images and 7K evaluation images. For both datasets, each image is accompanied by a corresponding language description.

### 4.1    Comparison with state-of-the-art methods

We make comparisons with language-based colorization methods (*e.g.*, LBIE [7], ML2018 [27], Xie2018 [49], L-CoDe [45], L-CoDer [5], and L-CoIns [6]) and automatic colorization methods

Table 2: User study results. ∗ marks our method receiving scarce-level descriptions. Ours (L-CAD) clearly produces a higher score than state-of-the-art methods on both datasets.

| Corresponding experiment | | | Reality experiment | | |
|---|---|---|---|---|---|
| Method | Extended COCO-Stuff | Multi-instance | Method | Extended COCO-Stuff | Multi-instance |
| LBIE | 1.68% | 3.20% | CIC | 6.24% | 3.76% |
| ML2018 | 3.28% | 4.92% | InstColor | 11.68% | 10.36% |
| Xie2018 | 0.96% | 1.72% | ChromaGAN | 7.16% | 6.32% |
| L-CoDe | 11.56% | 11.32% | BigColor | 11.48% | 9.56% |
| L-CoDer | 16.32% | 15.24% | DISCO | 8.92% | 10.84% |
| L-CoIns | 24.76% | 19.56% | $CT^2$ | 13.36% | 12.92% |
| Ours | **41.44%** | **44.04%** | Ours* | 19.72% | 20.28% |
| Ground truth | N/A | N/A | Ground truth | **21.44%** | **25.96%** |

(*e.g.*, CIC [52], InstColor [39], ChromaGAN [40], BigColor [20], DISCO [47], and $CT^2$ [44]) to demonstrate that our model is capable of effectively handling any-level descriptions.

**Qualitative comparisons.** Previous language-based colorization methods assume that users provide comprehensive color descriptions, resulting in suboptimal performance, particularly for objects without corresponding color descriptions. In contrast, leveraging the prior knowledge of Stable Diffusion [34] and our newly proposed instance-aware sampling strategy, our L-CAD presents vivid colorization results with any-level descriptions, as shown in Fig. 3. we further make comparisons with automatic colorization methods using scarce-level descriptions that lack meaningful color information. As shown in Fig. 4, our model still produces superior colorization results.

**Quantitative comparisons.** Following previous works [5, 6, 45], we use Peak Signal-to-Noise Ratio (PSNR) [16], Structural Similarity Index Measure (SSIM) [42], and Learned Perceptual Image Patch Similarity (LPIPS) [53] to qualitatively evaluate the quality of colorization. With complete-level and partial-level descriptions, our method (L-CAD) outperforms all language-based colorization methods. Furthermore, with scarce-level descriptions, our method shows a distinct advantage over automatic colorization methods. As shown in Tab. 1, our method achieves the best PSNR, SSIM, and LPIPS scores on both datasets.

## 4.2 User study

We further conduct user studies to evaluate the subjective perception of human observers: *(i)* Corresponding experiment: We apply language-based colorization methods to colorize images with random complete-level or partial-level descriptions. Participants are instructed to select the image that best matches the description. *(ii)* Reality experiment: Participants are asked to select the image they perceive to be the most visually realistic among the real image, images colorized by our method with scarce-level descriptions, and those colorized by automatic colorization methods. For each experiment, we randomly select 100 samples from the testing images of each dataset, and conduct them independently by 25 volunteers on the Amazon Mechanical Turk (AMT). As shown in Tab. 2, our method achieves the highest scores in both experiments.

## 4.3 Ablation study and discussion

We create three baselines to study the impact of our proposed modules and the sampling strategy. The evaluation scores and colorization results of the ablation study are shown in Tab. 1 and Fig. 5.

***W/o* Luminance-guided Image Compression (LIC).** We disable the luminance encoder in the pixel space and revert to using the perceptual compression model in [34]. Without the guidance of multi-scale grayscale features, the model cannot correctly preserve local spatial structures.

***W/o* Semantic-aligned Latent Representation (SLR).** We remove luminance features that guide the semantic alignment in the latent space and therefore replace channel-extended convolution blocks with vanilla convolution blocks. As a result, the obvious ghosting effect occurs in colorized images.

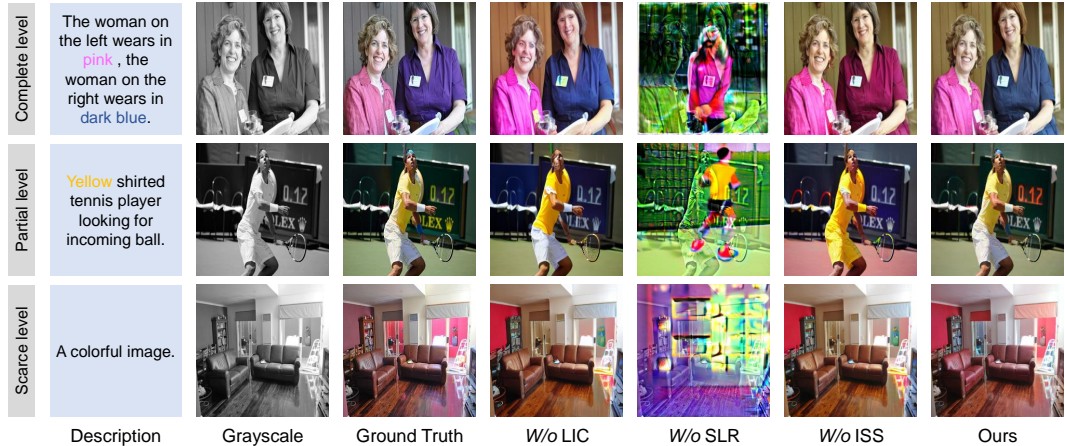

Figure 5: Ablation study. When our proposed modules are disable, results become counterintuitive.

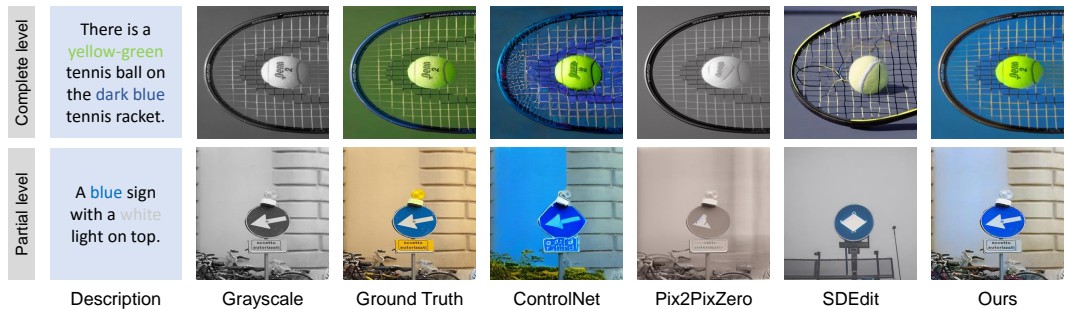

Figure 6: Compared with text-driven editing methods, our model colorizes images to be spatially aligned with grayscale images.

***W/o* Instance-aware Sampling Strategy (ISS).** We replace the instance-aware sampling strategy with the standard DDIM [38], which significantly degrades the performance of the model to correctly assign colors to corresponding objects that have different descriptions.

**Discussion.** Recent studies (*e.g.*, ControlNet [50], Pix2PixZero [30], and SDEdit [28]) could also edit images with descriptions, based on the pretrained generative models. However, these methods are not tailored to the colorization task, resulting in challenges related to preserving local spatial structures, leveraging rich color priors, and learning correspondence between objects and color words. These limitations make it difficult for them to perform instance-aware colorization with any-level descriptions. We present their unsuitability in Fig. 6.

### 4.4 Evaluation on ImageNet dataset

We conduct extensive experiments on the widely-used ImageNet dataset [36], including 1.3M images covering 1000 categories, to evaluate the performance of our L-CAD for automatic colorization. To provide a comprehensive evaluation, we introduce the Fréchet inception distance [12] to quantify the distribution gap between colorized images and ground truth images, as well as the colorfulness score [11] to reflect the vividness of colorized images. Following previous works [2, 22, 44], we evaluate our model on the first 5K images of the public validation set, where all the test images are center cropped and resized into $256 \times 256$ resolution.

As shown in Tab. 3, our model achieves top scores in FID, PSNR, SSIM, and LPIPS metrics, and demonstrates comparable performance in colorfulness metrics. These results indicate that our model aims to achieve photo-realistic colorization, emphasizing natural tones rather than exaggerated saturation. It should be noted that due to the absence of provided descriptions in the ImageNet

Table 3: Quantitative comparison with more automatic colorization methods on ImageNet dataset. ∗ marks our method receiving scarce-level descriptions.

| Method | FID↓ | PSNR↑ | SSIM↑ | LPIPS↓ | colorful↑ | △colorful↓ |
|--------|------|-------|-------|--------|-----------|------------|
| CIC [52] | 8.72 | 22.64 | 0.91 | 0.22 | 31.60 | 4.72 |
| DeOldify [1] | 9.45 | 21.12 | 0.83 | 0.24 | 22.70 | 13.62 |
| ChromaGAN [40] | 7.66 | 23.35 | 0.90 | 0.21 | 27.88 | 8.43 |
| InstColor [39] | 8.06 | 23.28 | 0.91 | 0.21 | 24.87 | 11.44 |
| GCP [46] | 5.95 | 21.68 | 0.88 | 0.23 | 32.98 | 3.34 |
| BigColor [20] | 10.08 | 21.35 | 0.87 | 0.25 | 30.60 | 7.12 |
| DSICO [47] | 7.99 | 20.31 | 0.86 | 0.23 | **48.43** | 10.70 |
| ColTran [22] | 6.44 | 20.95 | 0.80 | 0.29 | 34.50 | 2.24 |
| CT$^2$ [44] | 5.51 | 23.50 | 0.92 | 0.19 | 38.48 | 2.17 |
| ColorFormer [18] | 4.64 | 23.14 | 0.89 | 0.18 | 37.95 | **0.23** |
| L-CAD* | **4.36** | **24.47** | **0.92** | **0.16** | 34.04 | 3.68 |

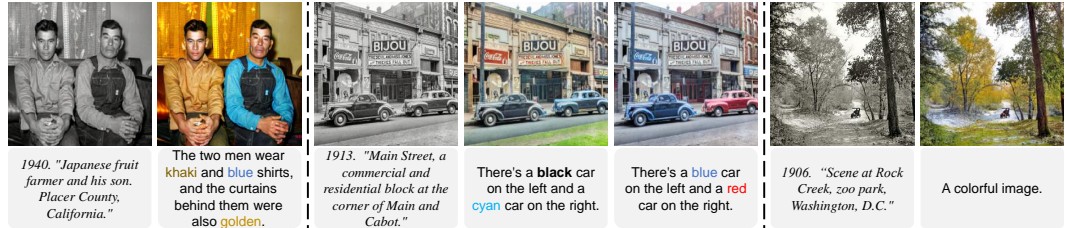

*1940. "Japanese fruit farmer and his son. Placer County, California."* | The two men wear khaki and blue shirts, and the curtains behind them were also golden. | *1913. "Main Street, a commercial and residential block at the corner of Main and Cabot."* | There's a **black** car on the left and a cyan car on the right. | There's a blue car on the left and a red car on the right. | *1906. "Scene at Rock Creek, zoo park, Washington, D.C."* | A colorful image.

Figure 7: Examples of applying our method to colorize legacy photos with any-level descriptions.

dataset [36], our model is trained solely with scarce-level descriptions (*e.g.*, "a colorful image") and leverage the rich image semantics encapsulated in language-based descriptions by initializing weights pretrained on the extended COCO-Stuff dataset [45]. This deviates from the training strategy adopted in language-based colorization datasets [6, 45] (detailed in Sec. 3.5), potentially affecting the performance of our model.

## 4.5   Application

We apply our method to colorize legacy black-and-white photos using descriptions at the complete, partial, and scarce levels, respectively. Results are presented sequentially from left to right in Fig. 7, which demonstrates the robust generalization capability of our model.

## 5   Conclusion

We present **L-CAD**, a unified model to perform **L**anguage-based **C**olorization with **A**ny-level **D**escriptions. Leveraging the prior knowledge of the pretrained model, we design novel modules to preserve local spatial structures and prevent the ghosting effect. we further propose the instance-aware sampling strategy to colorize diverse and complex scenarios. We demonstrate our model could handle any-level descriptions by making comparisons with language-based colorization methods with complete-level and partial-level descriptions, as well as automatic colorization methods with scare-level descriptions. Both qualitative and quantitative results demonstrate our superior performance.

**Limitations.** Our model as a diffusion model takes an additional sampling process for visualizing colorization results. This requirement considerably slows down the generation speed, and further limits the potential for real-time applications of our model. To address this concern, adopting advanced fast sampling methods (*e.g.*, DPM-Solver [26]) could be a promising direction.

**Acknowledgement.** This work is supported by the National Key R&D Program of China under Grant No. 2021ZD0109800, the National Natural Science Foundation of China under Grand No. 62136001, 62088102, and Program for Youth Innovative Research Team of BUPT No. 2023QNTD02. Yu Li is supported by Shenzhen Hetao Shenzhen-Hong Kong Science and Technology Innovation Cooperation Zone, under Grant No. HTHZQSWS-KCCYB-2023052.

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
