# Supplmentary Material:
# L-CAD: Language-based Colorization with Any-level Descriptions using Diffusion Priors

**Zheng Chang**[#1] **Shuchen Weng**[#2,3] **Peixuan Zhang**[1] **Yu Li**[4] **Si Li**[*1] **Boxin Shi**[2,3]

[1] School of Artificial Intelligence, Beijing University of Posts and Telecommunications
[2] National Key Laboratory for Multimedia Information Processing
School of Computer Science, Peking University
[3] National Engineering Research Center of Visual Technology
School of Computer Science, Peking University
[4] International Digital Economy Academy
{zhengchang98,pxzhang,lisi}@bupt.edu.cn
{shuchenweng, shiboxin}@pku.edu.cn    liyu@idea.edu.cn

## 6   Appendix

### 6.1   Robustness for contour estimation

We leverage a referring segmentation model to roughly estimate object contours mentioned in the description, which enables us to perform the instance-aware sampling strategy. To further demonstrate the robustness of our model, we manually annotate a sequence of contours ranging from coarse to fine and visualize the corresponding colorization results. As shown in Figure 8, our model presents a remarkable ability to produce condition-consistent colorization results even using imprecise contours. This is because the sampling is performed in the latent space using downsampled contours and the compression decoder in the pixel space could adaptively fix color bleeding issues.

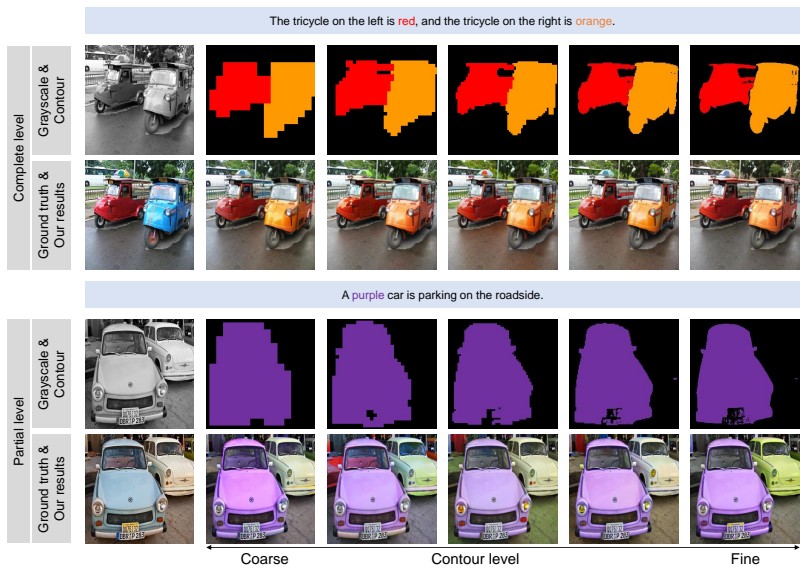

Figure 8: Visualization of colorization results by applying contours from coarse to fine.

---

# Equal contribution. * Corresponding author

37th Conference on Neural Information Processing Systems (NeurIPS 2023).

## 6.2 Additional comparison results

As presented in Sec. 4.1 of the main paper, we comprehensively evaluate our method on language-based colorization datsets, where we make comparisons with language-based colorization methods (*e.g.*, LBIE [3], ML2018 [6], Xie2018 [14], L-CoDe [12], L-CoDer [1], and L-CoIns [2]) using complete-level and partial level descriptions, and comparisons with automatic colorization methods (*e.g.*, CIC [15], InstColor [9], ChromaGAN [10], BigColor [5], DISCO [13], and $CT^2$ [11]) using scarce-level descriptions.

Following the evaluation protocol on ImageNet dataset [7], we evaluate colorization results at the more common resolution of $256 \times 256$, instead of $224 \times 224$ resolution in previous works [1, 2, 12]. This higher resolution increases the difficulty of the colorization, resulting in slightly lower scores for the quantitative metrics (see Tab. 1 of the main paper), compared to those reported in previous works [1, 2, 12]. Additionally, we provide more qualitative comparison results with language-based colorization methods and automatic colorization methods in Fig. 9 and Fig. 10, respectively.

## 6.3 Additional ablation results

To demonstrate the effectiveness of our proposed luminance-guided image compression, semantic-aligned latent representation, and instance-aware sampling strategy (details in Sec. 4.3 of the main paper), we create three baselines by disabling corresponding modules. Additional qualitative ablation study results are shown in Fig. 11.

## 6.4 Additional application results

We demonstrate our generalization capability by showing more colorization results on legacy black-and-white photos in Fig. 12, where results are presented sequentially from left to right using descriptions at the complete, partial, and scarce levels.

## 6.5 Diverse colorization results

By leveraging the inherent stochasticity of diffusion models [4, 8], which sample noise from a Gaussian distribution at each step of the denoising process, our method could effectively generate diverse colorization results for unmentioned objects in descriptions. We show our diverse colorization results with partial-level and scarce-level descriptions in Fig. 13.

Furthermore, we present more challenging results of our L-CAD using complete-level, partial-level, and scarce-level descriptions in Fig. 14, Fig. 15, and Fig. 16, respectively. These demonstrate that our method could produce high-quality colorization results for diverse and complex scenarios.

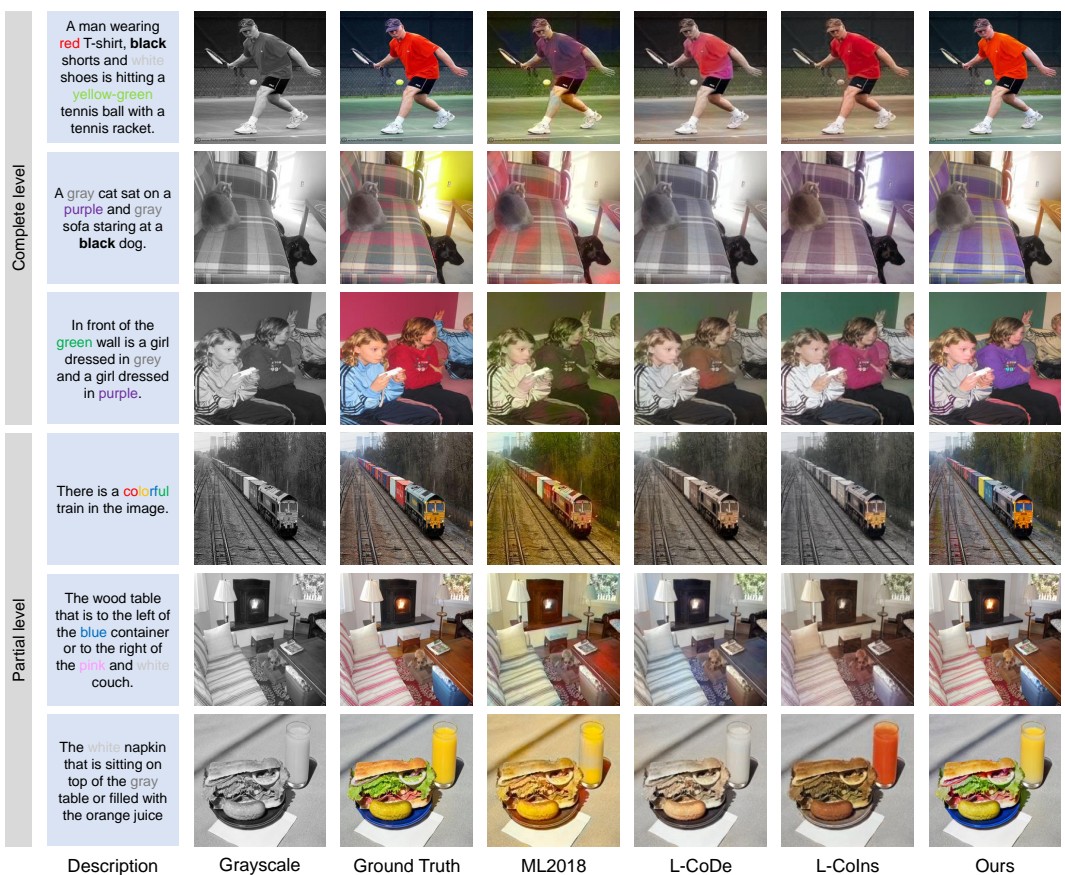

Figure 9: More comparison results with language-based colorization methods.

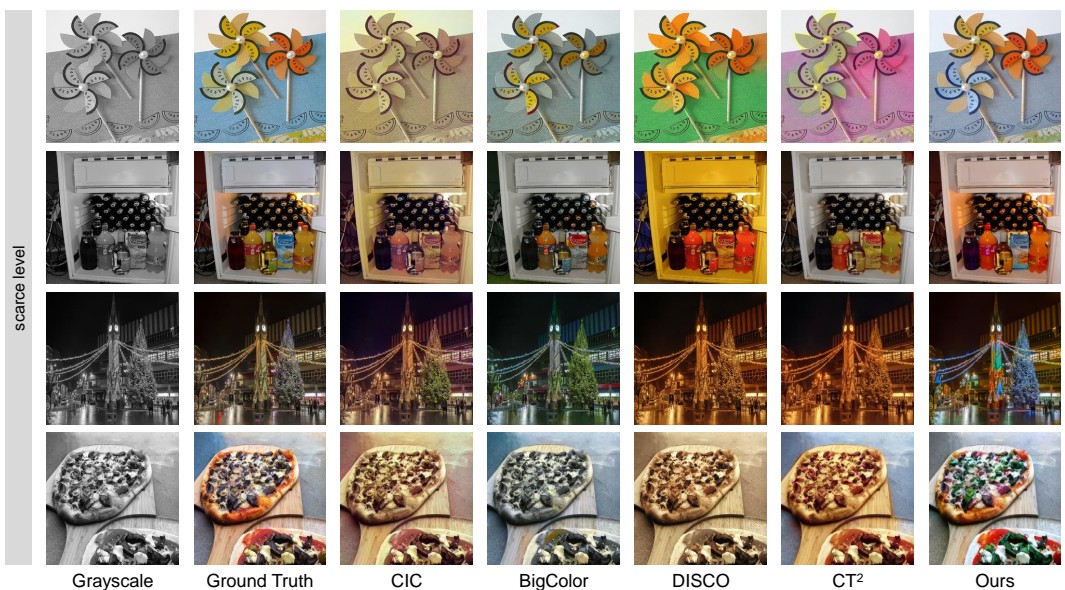

Figure 10: More comparison results with automatic colorization methods.

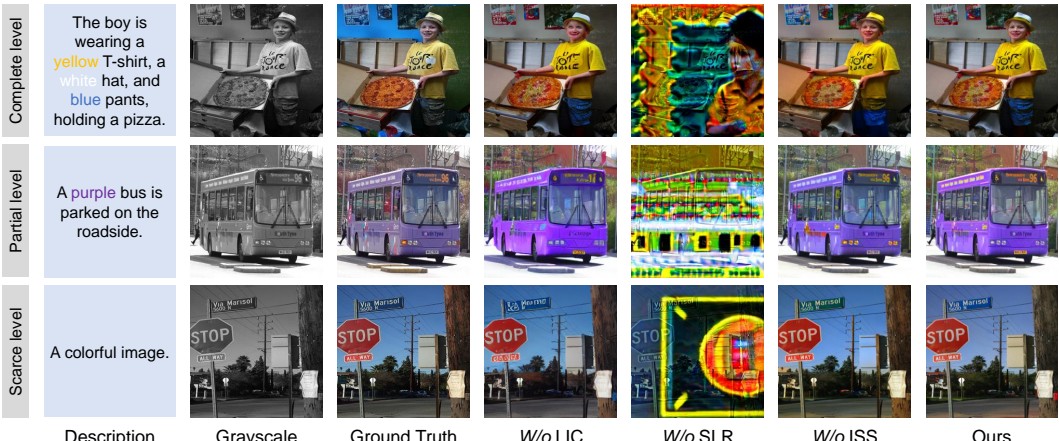

Figure 11: More ablation study results.

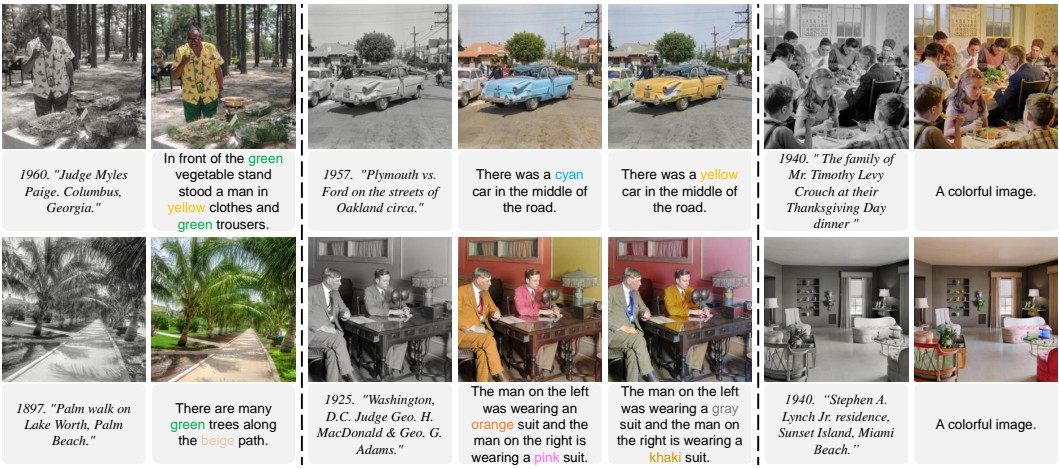

Figure 12: More colorization results of legacy black-and-white photos.

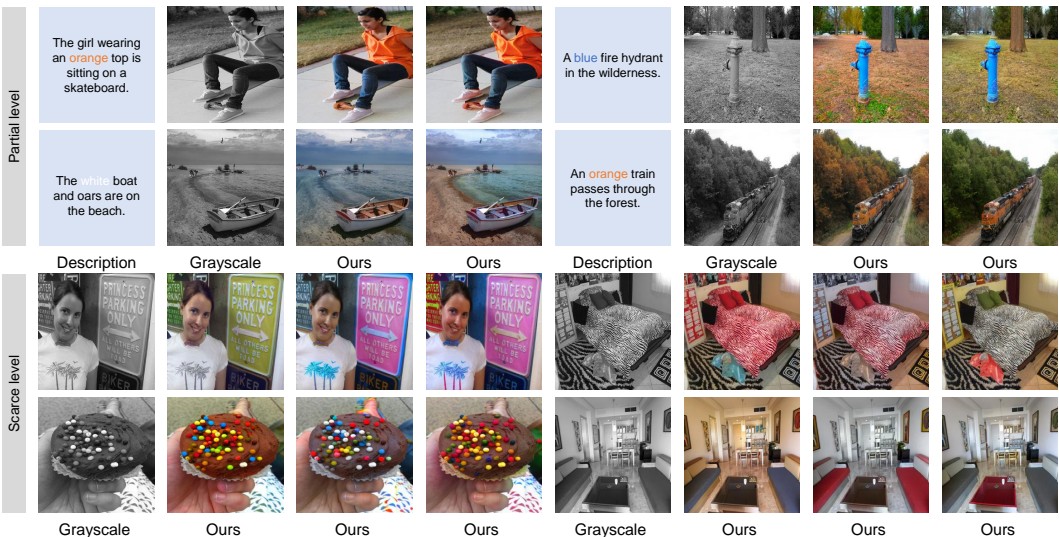

Figure 13: Diverse colorization results.

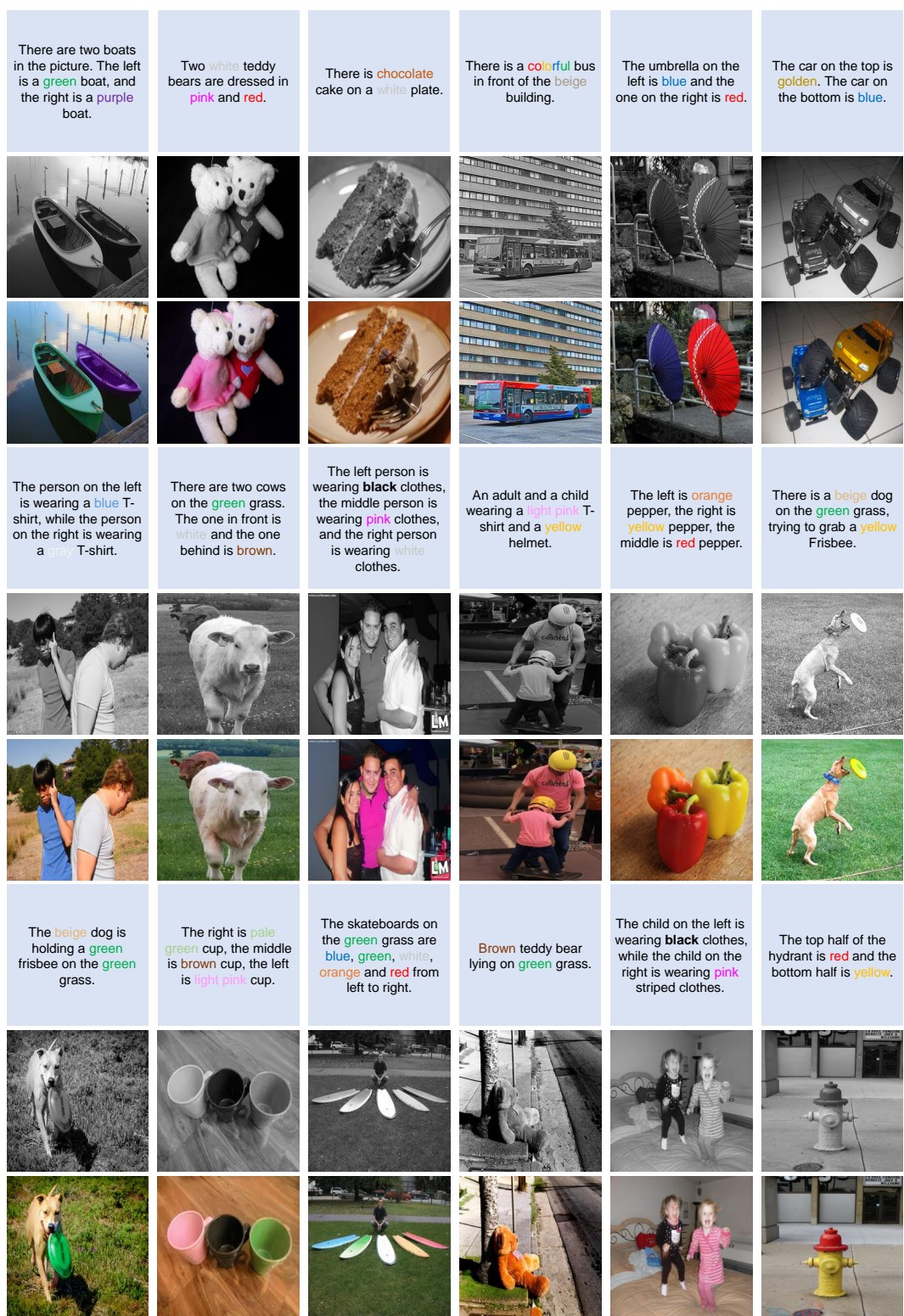

Figure 14: More results of our L-CAD using complete-level descriptions.

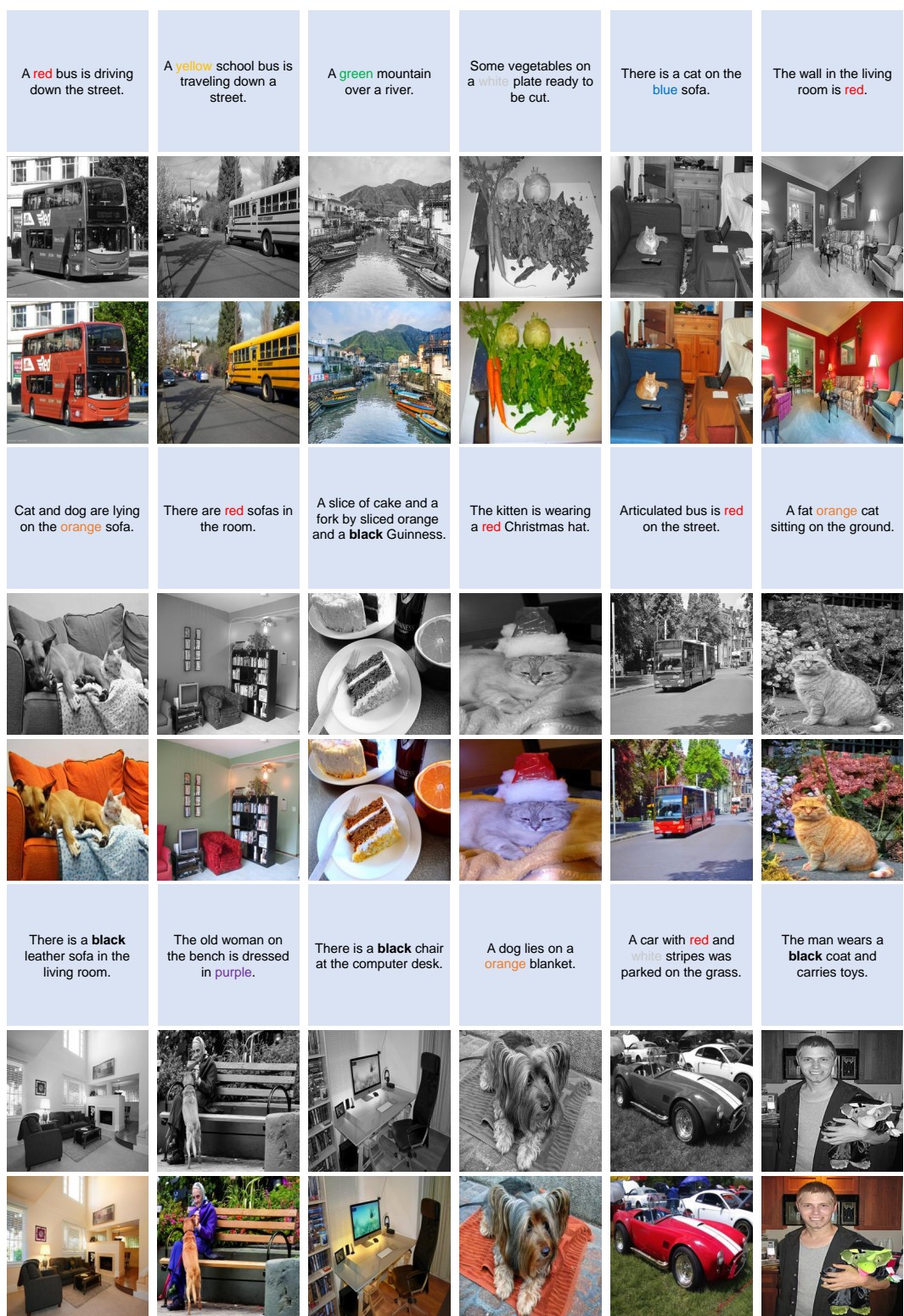

Figure 15: More results of our L-CAD using partial-level descriptions.

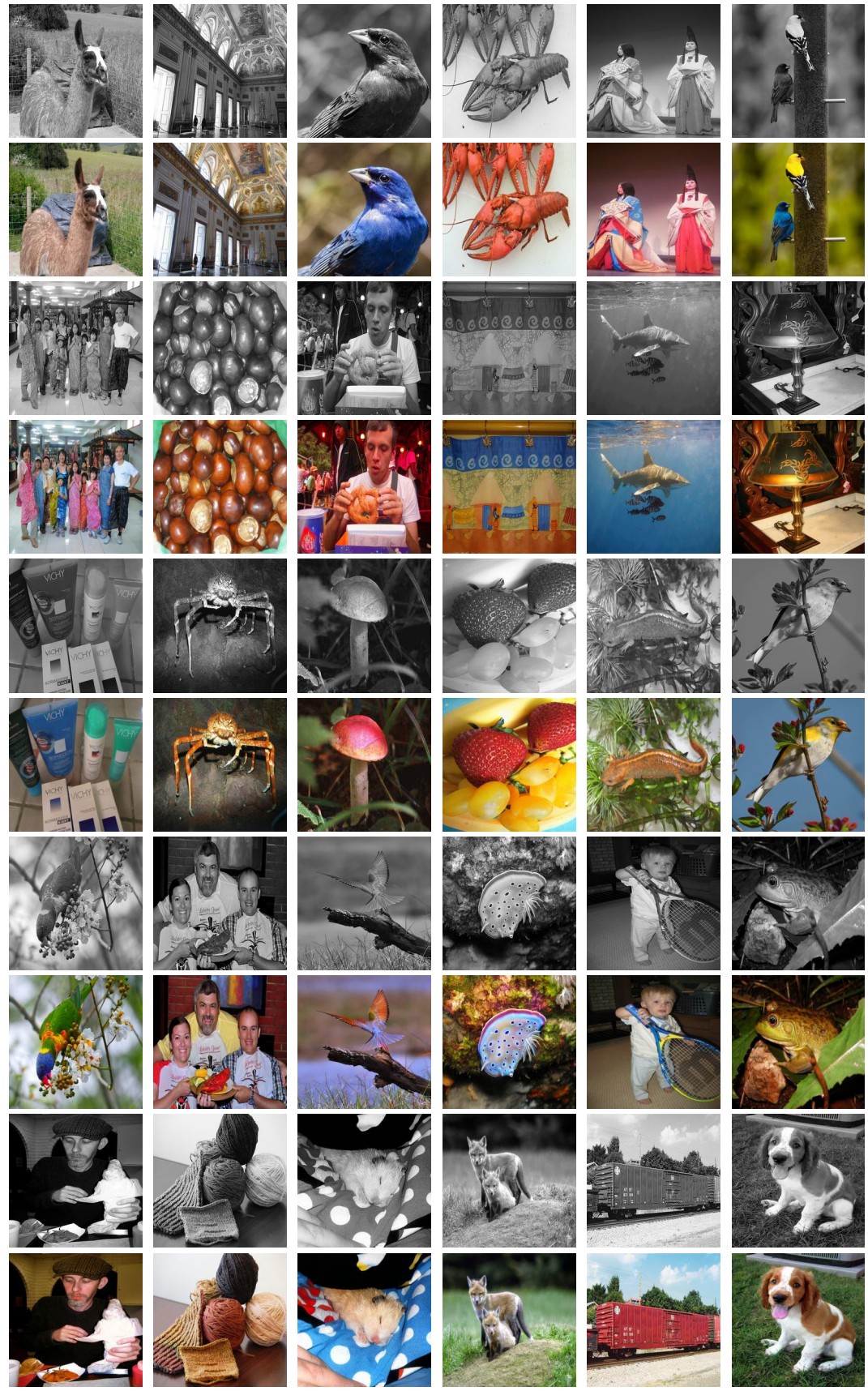

Figure 16: More results of our L-CAD using scarce-level descriptions.