# OpenReview forum: "L-CAD: Language-based Colorization with Any-level Descriptions using Diffusion Priors"
_NeurIPS.cc/2023/Conference — NeurIPS 2023 spotlight_

### Official Review · Reviewer_oFt4 · 2023-06-22

**Soundness:** 2 fair
**Presentation:** 2 fair
**Contribution:** 3 good
**Rating:** 7
**Confidence:** 4

**Summary:**

In this paper, the authors propose a model to adapt pretrained Stable Diffusion for language-conditioned colorization. In specific, they propose a luminance encoder that produces latents $y_lum$. This conditions the compression decoder and the denoiser of the diffusion model. The conditioning of the denoiser is done using a "Channel Extended Convolution block". Further during sampling, they utilize gradients from a semantic segmentation model to help with instance differentiation.

Comparisons are made to several language-conditioned and unconditional baselines across 2 datasets: extended coco-stuff and a (private) multi-instance dataset. They outperform these baselines on both quantitative metrics and mechanical turk. They show qualitative samples where they are able to colorize multi-level descriptions better than the baselines.

**Strengths:**

The results shown by their proposed approach are quite strong. The authors propose to augment a pretrained stable diffusion model to achieve language-conditoned colorizaiion which is novel and interesting. The authors provide adequate background information.

**Weaknesses:**

While the model works well, imho some novelty aspects are overclaimed and some simpler baselines are not ablated. I initially rate the paper slightly below borderline but I'm happy to update my rating it the authors address my concerns. See below for a detailed list of questions.

**Questions:**

**Most Important**: Amongst the standard training hyperparameters, the paper introduces $\lambda$ for sampling, the loss hyperparameters $\alpha$ and $\beta$ and the number of elements $N_{win}^2$ in the window for the loss in Eq 3). How are these hyperparameters tuned?


Luminance-guided image compression
-----------------------------------------

* The design of conditioning the “compression decoder” with the grayscale features makes sense. To be more self-conditioned, I suggest the authors to a) Briefly explain the architecture of the compression encoder + decoder b) Describe how exactly the conditioning from the luminance encoder features is incorporated in the compression decoder?
* In Section 3.2, the authors introduce a discriminator loss but this is not explained or ablated. Is this introduced in this work are part of stable diffusion?
* The authors introduce a loss that promotes smoothness by upweighting windows with high variance (Eq 3). The authors should ablate this weighting factor. Moreover, I think only the “luminance conditioning” is introduced in this work. So I suggest that the authors move the part that focuses on the training the compression encoder+decoder with the loss functions into the preliminaries and focus only on the luminance conditioning in this subsection.

Semantic-aligned latent representation
-------------------------

* Imho, the novelty of this section is overclaimed. The “semantic-aligned” latent representation amounts to a zero-initialized convolution which takes as input to the grayscale features and is added to the output of denoising downsampling blocks. It is great that it works, but the authors should describe it as it is and not overclaim novelty.
* Have the authors tried the obvious baseline that just concatenates $y_lum$ with $z_t$ at the inputs with downsampling to match the shapes?

Instance-aware sampling strategy
-----------------------

* IIUC, the goal is to differentiate between different instances and if the output of a semantic segmentation model is used to provide this information as some sort of guidance or ground truth. If this is indeed the case, then the authors should ablate the simpler baseline, which is just to condition the denoising process with the output of a semantic segmentation model during training (as they do with the grayscale images)
* What eactly is the semantic segmentation model used here? The authors say (eg SAM but it it not clear if SAM is used)
* The authors should mention the shapes of the $M_est$ and $M_att$ and how they are matched.
* What is the role of the softmax? Since it is cross-attention with the text tokens, aren’t the attention values normalized across the #text tokens already?

Results
---------
* Can the authors confirm that the images apart from the 2.9K descriptions that are removed, all have color information?
* The authors describe a multi-instance dataset. Are there plans to opensource this dataset?
* In Figure 1, the authors describe three levels of colorization descriptions. The authors show some qualitiative results per-level, can they report some quantitative results as well?
* Can the authors also report some of their failure cases, if any?

Other
-------
Figure 2 combines both the training and sampling loop of the diffusion process, so it is quite confusing. I suggest that the authors separate both and clearly label the diagram wherever the loss is applicable.

**Limitations:**

I do not believe there is negative societal impact of this work.

---

> ### Author Rebuttal · Authors · 2023-08-03
>
> Thanks for the very detailed review and suggestions.
>
> Given the character limit (6000), we have to make our response brief. For additional details, we welcome a more comprehensive discussion during the Author-Reviewer Discussions.
>
> Note Fig. S1-S13 and Tab. S1-S2 are included in the PDF attached to the global response.
>
> ### ***Most important***
> - Q: How are these hyperparameters tuned?
>
> We follow Stable Diffusion [31] to set loss hyperparameters $\alpha=1.0$ and $\beta=0.5$, and empirically set hyperparameters $\lambda=0.1$ and $N_\mathrm{win}=7$.
>
> We visualize artifact maps with varying $N_\mathrm{win}$ to illustrate its practical role, as shown in Fig. S2. We further modulate $\lambda$ and present qualitative and quantitative results in Fig. S9 and Tab. S1, respectively. This demonstrates that the value of $\lambda$ is not sensitive to variations in a specific range.
>
> ### ***Luminance-guided image compression***
> - Q: Explain the architecture of compression encoder/decoder and how luminance encoder features are incorporated.
>
> We implement the compression encoder and compression decoder as the same structure of Stable Diffusion [1], as shown in Fig. S1. As presented in Fig. 2 (a), luminance encoder extracts multi-scale features from grayscale images (L133-135). Then, these features are added to corresponding scales of the compression decoder (L135-136).
>
> - Q: Explain discriminator loss in Sec. 3.2.
>
> We adopt the exactly same discriminator loss as Stable Diffusion [1], therefore we do not perform an ablation study for it.
>
> - Q: Aablating this weighting factor in Eq. 3.
>
> We apply an additional factor $\gamma$ to adjust the weight of $\mathcal{L}_\mathrm{rec}$ when training our model in the pixel space. The qualitative and quantitative results are shown in Fig. S10 and Tab. S1, respectively. This demonstrates the value of $\gamma$ is also not sensitive to variations in a certain range.
>
> ### ***Semantic-aligned latent representation***
> - Q: Novelty overclaim in Sec. 3.3.
>
> Thanks for pointing this out. Although CEC block is mathematically equivalent to using a stack of convolutions to extract features and add their output to the downsampling block, we present it this way to underscore the core motivation and practical value of the module in establishing a semantically-aligned latent representation (See L159-160 and L165-168). We will revise this in the final version to tone down the claim of novelty, *e.g.*, removing ''novel'' in L157 and explaining the mathematical equivalence.
>
> - Q: The concatenation baseline.
>
> We conduct an additional ablation study ''concat''. Given the increase in the number of input channels, we extend the input channel of the first convolution layer. This modification leads to clear degradation in performance. We show the qualitative and quantitative results in Fig. S11 and Tab. S1, respectively.
>
> ### ***Instance-aware sampling strategy***
> - Q: Segmentation guided baseline.
>
> We build a ''segmentation'' baseline, where estimated object contours are concatenated with luminance features. The qualitative and quantitative results are shown in Fig. S11 and Tab. S1, respectively. This demonstrates that our instance-aware sampling strategy (ISS) offers a more effective control strategy.
>
> - Q: Which is semantic segmentation model used?
>
> We exactly use SAM [18] as the segmentation model when sampling colorization results. Note that our model could switch to an arbitrary segmentation model without finetuning with ISS.
>
> - Q: Given shape of $M^\mathrm{est}$ and $M^\mathrm{att}$
>
> The shape of $M^\mathrm{att}$ is $\bar{h} \times \bar{w} \times N_\mathrm{obj}$, where $\bar{h}$ and $\bar{w}$ are corresponding spatial resolution at $l$-th CA block, and $N_\mathrm{obj}$ is the number of objects in the description. The shape of $M^\mathrm{est}$ is $H \times W \times N_\mathrm{obj}$, where $H$ and $W$ are spatial resolutions of input grayscale image. we downsample its spatial resolution to $\bar{h} \times \bar{w}$ to ensure compatibility with $M^\mathrm{att}$.
>
> - Q: The role of the softmax.
>
> We apologize for typos in L193-194, which should be rewritten as:
>
> L193 $\mathcal{M} \leftarrow \mathrm{Sigmoid}(M^\mathrm{att}_l)$
>
> L194 $\hat{M}^\mathrm{att}_l \leftarrow M^\mathrm{att}_l - \lambda \nabla _\mathcal{M} \mathcal{L} _\mathrm{BCE}(\mathcal{M}, \hat{M}^\mathrm{est}_l)$
>
> We integrate L193-194 into the attention mechanism. Specifically, we execute matrix multiplication to compute unnormalized attention maps, and then modify them (L193-194) followed by a softmax operation. This procedure aligns $M^\mathrm{att}_l$ with downsampled estimated contours $\hat{M}^\mathrm{est}_l$.
>
> We will revise this in the final version.
>
> ### ***Result***
> - Q: Whether all data have color information.
>
> Previous L-CoDe [41] and L-CoIns [5] ensure all descriptions of the extended COCO-stuff and multi-instance dataset include color information.
>
> - Q: Release Multi-instance dataset.
>
> L-CoIns [5] has released the multi-instance dataset.
>
> - Q: Per-level qualitative results.
>
> There is no distinct criterion to partition descriptions of the extended COCO-stuff dataset and multi-instance datasets into complete-level and partial-level representations. As such, we present the quantitative results which include descriptions from both levels, as shown in Tab. 1 (left). We further show qualitative results with scarce-level descriptions in Tab. 1 (right) and Tab. 3 of the supplementary materials.
>
> - Q: Failure cases.
>
> Given that the estimated object contours are utilized solely in the latent space and subsequently become low-resolution, our method still encounters challenges when attempting to accurately colorize small objects with corresponding color descriptions. We provide failure cases in Fig. S12.
>
> ### ***Other***
> - Q: Confused Fig. 2.
>
> Thanks for the helpful suggestion. We have redesigned Fig. S13 to serve as the revised Fig. 2, enhancing the clarity of our pipeline.
>
> We will revise this in the final version.

---

> > ### Comment · Reviewer_oFt4 · 2023-08-15
> > **Updated Rating**
> >
> > Thanks to the authors for the new experiments. I updated my rating to 7.
> >
> > Can the authors explain Fig. S2 more? Why is $N_{win} = 7$ optimal?

---

> > > ### Author Response · Authors · 2023-08-16
> > > **Explaination of Fig. S2**
> > >
> > > Thanks for your comments.
> > >
> > > As shown in Fig. S2, we observe that a smaller $N_\mathrm{win}$ provides fewer, yet more accurate, artifactual positions. In contrast, a larger $N_\mathrm{win}$ presents a higher number of these positions, but at the expense of accuracy. To strike a balance, our approach is designed to maintain a reasonable level of accuracy while ensuring a sufficient number of evident artifactual positions.

---

### Official Review · Reviewer_ZuAx · 2023-07-05

**Soundness:** 3 good
**Presentation:** 2 fair
**Contribution:** 3 good
**Rating:** 6
**Confidence:** 2

**Summary:**

The paper proposes a modification over Stable Diffusion to adapt it to perform language-based colorization of images with varying levels of description details.
The method is based on three modifications. First, in addition to the auto-encoder used for SD, the authors add another encoder that is tasked with preserving the spatial features of the input image. Second, the convolutions in the downsampling layers of the UNet are replaced with a novel Channel-Extended Convolution to encourage the reliance on the encoded spatial features. Finally, a segmentation model is employed to encourage correct object-level color assignment.
Extensive experiments are conducted to demonstrate the method's superiority over both language-conditioned colorization methods and automatic colorization methods.

**Strengths:**

- The authors conduct very extensive experiments, using various datasets and human evaluation.
- Overall, the reviewer found the qualitative results to be convincing.
- The idea of slightly modifying the Stable Diffusion architecture for other tasks is interesting and can possibly be generalized to other tasks.

**Weaknesses:**

Please note that the low confidence is due to the fact that the reviewer is not familiar with literature on image colorization, and therefore feels less confident in giving an assessment of this work.

Readability:
Overall, the reviewer found the method section to be a bit hard to follow. Intuitions are lacking in some parts, and familiarity with previous works is required to follow the explanations.
- The writing of the preliminaries section on diffusion models is a bit confusing and inaccurate. For example, it starts with describing the inference (L. 106-107) but then turns to describe the training process (Eq. 1 and on).
- Section 3.3 (Channel-Extended Convolution) is not entirely clear. A supporting figure demonstrating the VC vs. CEC would be beneficial to better understand the module, its motivation and novelty.
- Section 3.4 heavily relies on familiarity with existing colorization methods, and the manipulation to the cross-attention maps is not clear to the reviewer (e.g., why employ a sigmoid non-linearity on the attention?)

Evaluation:
- The authors claim that when ablating the Instance-aware Sampling Strategy (ISS) "significantly degrades the performance of the model to correctly assign colors to corresponding objects that have different descriptions." (L.256) however, the results in Fig. 5 and Tab. 1 appear to be pretty similar to the proposed method.
- The comparison to image editing methods are partial, however given the extensive comparisons, this is not a major concern.


**Questions:**

- The authors mention the "ghosting effect" several times, but no explanation is given as to what it is (e.g., L. 153).
- Why is the CEC block only used in the first half of the UNet (downsampling)?
- How robust is the method to OOD outputs? For example, what happens if you train on COCO-Stuff and evaluate on multi-instance?

**Limitations:**

Limitations are discussed briefly. The reviewer thinks that a more in-depth comparison of runtime to SD and to the baselines is needed.

---

> ### Author Rebuttal · Authors · 2023-08-03
>
> Thank you for your careful review and valuable feedback.
>
> Given the character limit (6000), we have to make our response brief. For additional details, we welcome a more comprehensive discussion during the Author-Reviewer Discussions.
>
> As the rebuttal instruction, Fig. S1-S13 and Tab. S1-S2 are shown in the PDF attached to the global response.
>
> ### ***Readability***
> - Q: Confused preliminaries.
>
> Thanks for the suggestions. To enhance the clarity and comprehensibility of the preliminaries section, we will incorporate additional background information and strategically reorganize the arrangement of the paragraphs in the final version.
>
> - Q: Figure about VC vs. CEC.
>
> The CEC block is designed to extend the channel number of vanilla convolution block so that the model could utilize the extended channels to effectively capture the local structural semantics of the luminance in the latent space (L159-160). By initializing the weights of extended channels to zero, CEC block ensures that our model maintains functional equivalence to the pretrained generative model prior to training (L166-168).
>
> We provide a figure demonstrating the VC vs. CEC in Fig. S6 for better clarity.  In application, we feed the concatenation of resized luminance features and feature maps into our CEC block to extract joint feature maps.
>
> - Q: Relying on existing methods.
>
> Previous works assume that users provide comprehensive color descriptions for most of the objects in the image, which causes suboptimal performance (L25-26). Given the inherent ambiguity in the number of objects mentioned in any-level descriptions (L38-39), we leverage the pretrained cross-modality generative model (*i.e.*, Stable Diffusion [31]) to utilize its robust language understanding for mentioned objects and rich color priors for unmentioned ones (L39-41).
>
> - Q: Cross-attention maps.
>
> We apologize for typos in L193-194, which should be rewritten as:
>
> L193 $\mathcal{M} \leftarrow \mathrm{Sigmoid}(M^\mathrm{att}_l)$
>
> L194  $\hat{M}^\mathrm{att}_l \leftarrow M^\mathrm{att}_l - \lambda \nabla _\mathcal{M} \mathcal{L} _\mathrm{BCE}(\mathcal{M}, \hat{M}^\mathrm{est}_l)$
>
> We briefly introduce the manipulation procedures applied to attention maps: *(i)* Matrix multiplication is firstly utilized to compute the unnormalized attention maps $M_l^\mathrm{att} \in \mathbb{R}^{\bar{h} \times \bar{w} \times N_\mathrm{obj}}$, where $\bar{h}$ and $\bar{w}$ are corresponding spatial resolution at $l$-th CA block, and $N_\mathrm{obj}$ is the number of objects in the description. *(ii)* Then, we apply L193-194 to manipulate the attention maps $M^\mathrm{att}_l$, utilizing sigmoid function to confine the value range of attention maps to $[0,1]$ — the same value range as that of the estimated object contours $M^\mathrm{est}_l$. *(iii)* Finally, we apply a softmax operation to normalize the modified $\hat{M}^\mathrm{att}_l$.
>
> We will revise this in the final version.
>
> ### ***Evaluation***
> - Q: Effectiveness of Instance-aware Sampling Strategy (ISS).
>
> The performance improvement by adopting ISS is notable.
>
> As presented in Tab. 1 (left), when provided with complete-level and partial-level descriptions, the absence of ISS (*W/o* ISS) results in a decline in PSNR from 25.97 to 25.32 (-0.65) on the extended COCO-stuff dataset. Furthermore, for the multi-instance dataset, which provides samples featuring multiple instances with different visual characteristics within a single image, the PSNR drop becomes larger from 25.51 to 24.57 (-0.94). Figure 5 reveals that *W/o* ISS may not correctly assign colors to corresponding objects that have different descriptions, *e.g.*,  the right woman in the first row being incorrectly colorized in pink instead of dark blue, and the man's white pants in the second row being colorized as yellow.
>
> It is only in the context of scarce-level descriptions, which inherently lack meaningful color information for objects, that the model without ISS exhibits comparable performance. In these scenarios, the significance of ISS understandably diminishes.
>
> - Q: Comparison to image editing methods are partial.
>
> We appreciate the reviewer's understanding. The primary objective of this study addresses persistent challenges in colorization task, *i.e.*,  colorizing images with descriptions of varying detail levels.
>
> ### ***Questions***
> - Q: Ghosting effect.
>
> This concept is recognized in the field of photography and image restoration [R1]. In the context of our work, the ''ghosting effect'' implies the model synthesizes an image resembling a composite created from multiple blended images. *E.g.*, in the second row of Fig. 5, *W/o* SLR appears to produce an image of a man in a yellow shirt and subsequently merge it with the original grayscale image. We will explain this term clearly in the final version.
>
> - Q: CEC block in downsampling.
>
> We strategically apply CEC block in downsampling modules to expedite training convergence. Additionally, we conduct an ablation study where we replace VC blocks in both upsampling and downsampling modules with CEC blocks, denoted as ''upsampling''. The qualitative and quantitative results are shown in Fig. S7 and Tab. S1, respectively. Consequently, this ablation results in comparable performance.
>
> - Q: Robustness of OOD.
>
> We train models on extended COCO-stuff dataset and subsequently evaluate it on the multi-instance dataset, and vice versa. We name this experiment as ''OOD'', and provide qualitative and quantitative results in Fig. S8 and Tab. S1, respectively. These results demonstrate a certain degree of robustness of our method.
>
> ### ***Limitations***
> - Q: In-depth comparison of runtime to baselines.
>
> We include training and inference time for all methods in Tab. S2, which indicates our method cost more inference time.
>
> This could be mitigated by using advanced fast sampling methods (DPM-Solver++).
>
> [R1] YC Shih, D Krishnan, F Durand, and WT Freeman. Reflection removal using ghosting cues. In *CVPR* 2015.

---

### Official Review · Reviewer_whze · 2023-07-07

**Soundness:** 3 good
**Presentation:** 2 fair
**Contribution:** 3 good
**Rating:** 7
**Confidence:** 3

**Summary:**

This paper presents a novel approach to image colorization, which demonstrates superior performance in language-based colorization methods. The proposed model employs language descriptions at varying detail levels to produce high-quality, customizable colorized images by diffusion in the latent space. A key innovation is L-CAD's adaptive understanding of any-level descriptions, facilitating precise colorization based on user requests. To ensure proper spatial alignment with grayscale inputs and avoid ghosting effects, a luminance-guided compression module, and a channel-extended convolution operator are introduced. An instance-aware sampling strategy was adopted from previous literature to enhance color assignment to objects. L-CAD exhibits very good results in both quantitative and qualitatve experiments.

**Strengths:**

- The paper is well-written and easy to follow.
- The problem that they tackle is interesting and valuable.
- The results are promising and the ablation studies to disentangle different modules have been conducted.

**Weaknesses:**

- The method is a little complex and needs multiple modules to work properly.


**Questions:**

- Can the authors please compare training and inference time between their method and the baselines?

**Limitations:**

- The authors have mentioned that their model is relatively slow.

---

> ### Author Rebuttal · Authors · 2023-08-03
>
> Thank you for your positive feedback and suggestions.
>
> As the rebuttal instruction, Fig. S1-S13 and Tab. S1-S2 are shown in the PDF attached to the global response.
>
> ### ***Weaknesses***
> - Q: The method is a little complex.
>
> Previous works [4,5,6,24,41] implicitly assume that users provide comprehensive color descriptions for most of the objects in the image, which often leads to suboptimal performance, especially for objects without corresponding color descriptions (L24-26). To address the aforementioned issue, we intend to utilize Stable Diffusion [31]'s robust language understanding for mentioned objects and rich color priors for unmentioned ones (L38-41). Therefore, the model needs to be elaborately designed to be aligned with Stable Diffusion in the pixel space, the latent space, and the sampling strategy.
>
> As illustrated in Sec. 4.3, we demonstrate that every component of our model is indispensable for effectively performing language-based colorization with any-level descriptions.
>
> ### ***Questions***
> - Q: Training and inference time.
>
> Thanks for the suggestions, we present the training and inference time of all methods in Tab. S2.
>
> ### ***Limitations***
> - Q: Model is relatively slow.
>
> As illustrated in L278-279, this limitation could be mitigated by using advanced fast sampling methods (*i.e.*, DPM-Solver++ [R1]).  We show qualitative and quantitative results using DPM-Solver++ in Fig. S5 and Tab. S1 of the attached PDF, respectively. The accelerated inference time is presented in Tab. S2.
>
> [R1] C. Lu, Y. Zhou, F. Bao, J. Chen, C. Li, and J. Zhu. DPM-Solver++: Fast solver for guided sampling of diffusion probabilistic models. arXiv preprint arXiv:2211.01095, 2023.

---

### Official Review · Reviewer_Gyr2 · 2023-07-07

**Soundness:** 3 good
**Presentation:** 3 good
**Contribution:** 3 good
**Rating:** 7
**Confidence:** 4

**Summary:**

This paper presents a well engineered method for Stable
Diffusion-based text-guided image colorization. The paper addresses
the challenge of structural fidelity to the original grayscale image
by additional conditioning modules in the denoiser architecture. The
paper also proposes an instance-aware sampling procedure that uses
object segmentations to encourage accurate color assignment. Extensive
quantitative and qualitative comparison show the proposed method
outperforms the compared baselines.


**Strengths:**

- This is a well engineered method. The design decisions appear
  reasonable and are justified with ablation studies.
- The paper is well written, the ideas are clear and easy to understand.
- The qualitative results are remarkable and quantitative comparisons
  show the advantage of the proposed method in standard benchmarks and
  user preference studies.


**Weaknesses:**

- The colorization baselines in Figure 1 and 4 look much weaker than
  in the original papers (e.g. [43] Fig. 6 and [40] Fig. 5). I wonder
  how the proposed method would compare in the referenced figures.
- It would be interesting to include in the quantitative comparison
  the number of sampling steps required for each method, as this is a
  disadvantage of the diffusion-based methods compared to the GAN-based
  ones.


**Questions:**

See Weaknesses.

**Limitations:**

Yes.

---

> ### Author Rebuttal · Authors · 2023-08-02
>
> We thank the reviewer for this thoughtful review and we are glad to see their positive assessment.
>
> As the rebuttal instruction, Fig. S1-S13 and Tab. S1-S2 are shown in the PDF attached to the global response.
>
> ### ***Weaknesses***
> - Q: Colorization baselines look much weaker.
>
> For the performance assessment of [40] and [43], we adopt the publicly available pretrained weights (*i.e.*, [https://github.com/MenghanXia/DisentangledColorization](https://github.com/MenghanXia/DisentangledColorization) and [https://github.com/shuchenweng/CT2](https://github.com/shuchenweng/CT2)), which ensures a transparent and objective comparison.
>
> The scenarios displayed in Fig. 1 and Fig. 4 are particularly challenging due to the abundance of small objects with intricate textures.  In our observations, when objects in grayscale images present recognition difficulties, [40] and [43] tend to produce undersaturated colorization results. Leveraging the robust language understanding and rich color priors of Stable Diffusion [31], our model could adeptly achieve vivid colorization, even in such challenging cases. To further show the superior performance of our proposed method, we present supplementary qualitative experiments against [40] and [43] in typical outdoor scenarios where they have been known to perform satisfactorily. Please refer to Fig. S4.
>
> - Q: The number of sampling steps.
>
> It is noteworthy to point out that among the methods enumerated in Tab. 1 and Tab. 3, ours is the only one based on diffusion models.
>
> Therefore, we conduct an additional ablation study focusing on the impact of varying the number of sampling steps $N_\mathrm{step}$.  The corresponding evaluation scores and colorization results are presented in Tab. S1 and Fig. S5, respectively.
>
> Furthermore, we demonstrate that our method could be accelerated by adopting advanced fast sampling methods (*i.e.*, DPM-Solver++ [R1]) while maintaining high-quality colorization results, as shown in Fig. S5.
>
> [R1] C. Lu, Y. Zhou, F. Bao, J. Chen, C. Li, and J. Zhu. DPM-solver++: Fast solver for guided sampling of diffusion probabilistic models. arXiv preprint arXiv:2211.01095, 2023.

---

### Official Review · Reviewer_Gu3m · 2023-07-08

**Soundness:** 3 good
**Presentation:** 3 good
**Contribution:** 3 good
**Rating:** 6
**Confidence:** 4

**Summary:**

The paper addresses the problem of colorizing images with descriptions of diverse levels of details. The key idea of the work is to propose a unified model that adaptively understands any-level descriptions by leveraging the pretrained cross-modality generative model. Additionally, the paper introduces modules that aid in preserving local spatial structures and prevent the ghosting effect by aligning with input conditions in both the pixel space and the latent space. Further, the paper presents an instance-aware sampling strategy to correctly assign colors to corresponding objects, enabling effective colorization in diverse and complex scenarios. The work demonstrates state-of-the-art performance in both automatic and language-based colorization methods.

**Strengths:**

The paper is well-written and well-motivated. Qualitative and quantitative comparisons to existing methods show the effectiveness of the proposed approach in language based colorization and automatic colorization. User studies and ablation studies sufficiently justify the key conclusions made in the paper.


**Weaknesses:**

One of the key weaknesses of the paper is the readability of a few sub-sections. For example, I struggle to understand the detailed implementation/intuition for section 3.2 where the paper introduces luminance guided image compression and how it helps preserve local structural semantics. It is not very intuitive as to how it can help bridge alignment between colorization results and grayscale images. Some more detailed intuitions could help readers to better follow this section. Similarly the justifications for equation 3 could be further elaborated. In lines 153-154, on the ghosting effects, I am curious to know if this is still the case with using region based guidance (for example, see below two references). I will revise my scores based on the clarifications in the rebuttal.

Yang, Z., Wang, J., Gan, Z., Li, L., Lin, K., Wu, C., ... & Wang, L. (2023). Reco: Region-controlled text-to-image generation. In Proceedings of the IEEE/CVF Conference on Computer Vision and Pattern Recognition (pp. 14246-14255).

Li, Y., Liu, H., Wu, Q., Mu, F., Yang, J., Gao, J., ... & Lee, Y. J. (2023). Gligen: Open-set grounded text-to-image generation. In Proceedings of the IEEE/CVF Conference on Computer Vision and Pattern Recognition (pp. 22511-22521).


**Questions:**

Please see weaknesses.

**Limitations:**

Authors adequately addressed the limitations.

---

> ### Author Rebuttal · Authors · 2023-08-03
>
> We thank the reviewers for their insightful and constructive feedback.
>
> As the rebuttal instruction, Fig. S1-S13 and Tab. S1-S2 are shown in the PDF attached to the global response.
>
> ### ***Weaknesses***
> - Q: Detailed implementation/intuition for Sec. 3.2.
>
> Our primary intuition is to leverage Stable Diffusion [31]’s robust language understanding and rich color priors for language-based colorization with any-level descriptions (L125-126).  To realize this goal, we analyze the architecture of Stable Diffusion. This method adopts a compression encoder $\mathcal{E}$ to encode image $x$ from pixel space into latent space as $z = \mathcal{E}(x)$, and a compression decoder $\mathcal{D}$ to reconstruct the image as $\tilde{x}=\mathcal{D}(z)$ (L118-122). A critical observation is that Stable Diffusion lacks the ability to preserve local spatial structures of input grayscale images (L128-129). To remedy this limitation, we propose an additional luminance encoder $\hat{\mathcal{E}}$ in the pixel space as a bridge to align colorization results with grayscale images (L130-132). As shown in Fig. 2 (a), luminance encoder extracts multi-scale features from grayscale images, which preserves local structural semantics of grayscale images $\hat{\mathcal{E}}(x^{\mathrm{lum}})$ (L133-135). These features are added directly into the compression decoder, guiding its decoding process (L135-136).
>
> We implement the architecture of luminance encoder $\hat{\mathcal{E}}$ mirrors that of the compression encoder $\mathcal{E}$ of Stable Diffusion. The weights of compression encoder $\mathcal{E}$ and compression decoder $\mathcal{D}$ are fixed to retain prior knowledge from the pretrained model (L136-138). A visualization of the architecture for both the compression encoder and decoder is provided in Fig. S1.
>
> We will revise this in the final version for better readability.
>
> We will release the code, offering a more comprehensive understanding once the paper is accepted.
>
> - Q: Equation 3 could be further elaborated.
>
> When training the luminance encoder $\hat{\mathcal{E}}$, our goal is to identify and minimize the discrepancy between colorization results and grayscale images (L140-141). Since we observe that erroneous pixels significantly damage visual perception (L139-140), we intend to estimate an artifact map $M^\mathrm{art}_{h,w}$. This map would serve to indicate the probability of encountering artifacts at specific spatial position $(h,w)$ within the colorized results $\tilde{x}$. Specifically, we calculate the residual between the ground truth image and the colorized result as $\delta = x - \tilde{x}$ (L142-143). Next, we compute the variance of the aforementioned residual within local square windows at each position (L143-144),  as shown in Eq. 3.  Given that artifacts typically show up with high-frequency characteristics, areas with higher variances likely indicate where these artifacts are. Finally, we apply the artifact map as a weight to the image reconstruction loss (L146-147) to focus on the pixels where the model needs to further minimize the discrepancy.
>
> We further visualize the artifact map with different $N_{\mathrm{win}}$ to demonstrate the effectiveness of estimated artifacts map, as shown in Fig. S2.
>
> We will revise this in the final version for better readability.
>
> - Q: Using region-based guidance.
>
> To investigate whether object boxes could effectively mitigate the ghosting effects, we conduct two additional ablation studies by replacing modules in the latent space with the corresponding components from reference [R1] and [R2]. These modifications are individually designated as "L-ReCo" and "L-GLIGEN". To evaluate these ablation studies on the extended COCO-Stuff and Multi-instance datasets, we employ DINOv2 [R3] to estimate object boxes. While the results indicate a reduction in ghosting effects, they remain visible. This is because object boxes could only offer a coarse-grained location of primary objects, which is far less precise compared to the fine-grained luminance features provided by grayscale images. We show qualitative and quantitative results in Fig. S3 and Tab. S1, respectively.
>
> [R1] Z Yang, J Wang, Z Gan, L Li, K Lin, C Wu, N Duan, Z Liu, C Liu, M Zeng, and L Wang. ReCo: Region-controlled text-to-image generation. In *CVPR*, 2023 .
>
> [R2] Y Li, H Liu, Q Wu, F Mu, J Yang, J Gao, C Li, and YJ Lee. GLIGEN: Open-set grounded text-to-image generation. In *CVPR*, 2023.
>
> [R3] M. Oquab, T. Darcet, T. Moutakanni, H. Vo, M. Szafraniec, V. Khalidov, P. Fernandez, D. Haziza, F. Massa, A. El-Nouby,
> et al. DINOv2: Learning robust visual features without supervision. *arXiv preprint arXiv:2304.07193*, 2023.

---

> > ### Comment · Reviewer_Gu3m · 2023-08-15
> >
> > Thanks for clarifying my questions. I updated my review.

---

### Author Rebuttal · Authors · 2023-08-05

We thank the reviewers for their insightful comments and acknowledging that the paper is well-written (Gu3m/Gyr2/whze), well-motivated (Gu3m), and easy to follow (Gyr2/whze); proposed method is effective (Gu3m), reasonable (Gyr2), interesting and can be generalized (zuAx), and novel and interesting (oFt4); authors conduct extensive ablation or even human evaluation experiments (Gu3m/Gyr2/whze/zuAx); results are advantaged (Gyr2), promising (whze), convincing (zuAx), and quite strong (oFt4); tackled problem is interesting and valuable (whze); and background information is adequate (oFt4). We have carefully considered your comments and will take them into account to further improve the quality of our work. Please find below our responses to specific concerns of each individual reviewer.

Note that Fig. S1-S13 and Tab. S1-S2 can be found in the PDF attached to the global response. Once our paper is accepted, we will release the code and checkpoints to facilitate reproducibility and further research in this area.

We remain committed to addressing any further questions or concerns from the reviewers promptly.

---

### Decision · Program_Chairs · 2023-09-21

**Decision:**

Accept (spotlight)

**Comment:**

All the reviewers recommend the acceptance of the work. Reviewers appreciated the well-motivated and well-engineered method with good results. Multiple reviewers raised concerns with respect to readability and clarity of several parts of the paper. Authors addressed several of these concerns in the rebuttal and reviewers are satisfied with their responses. The reviewers did raise some valuable concerns that should be addressed in the final camera-ready version of the paper, which include adding the relevant rebuttal discussions and revisions in the main paper. The authors are encouraged to make the necessary changes to the best of their ability.